

# The influence of mortality and socioeconomic status on risk and delayed rewards: a replication with British participants

Gillian V. Pepper[1], D Helen Corby[2], Rachel Bamber[1], Holly Smith[1], Nicky Wong[1] and Daniel Nettle[1]

[1] Newcastle University, Newcastle Upon Tyne, Tyne and Wear, United Kingdom
[2] University of Edinburgh, Edinburgh, United Kingdom

## ABSTRACT

Here, we report three attempts to replicate a finding from an influential psychological study (*Griskevicius et al., 2011b*). The original study found interactions between childhood SES and experimental mortality-priming condition in predicting risk acceptance and delay discounting outcomes. The original study used US student samples. We used British university students (replication 1) and British online samples (replications 2 and 3) with a modified version of the original priming material, which was tailored to make it more credible to a British audience. We did not replicate the interaction between childhood SES and mortality-priming condition in any of our three experiments. The only consistent trend of note was an interaction between sex and priming condition for delay discounting. We note that psychological priming effects are considered fragile and often fail to replicate. Our failure to replicate the original finding could be due to demographic differences in study participants, alterations made to the prime, or other study limitations. However, it is also possible that the previously reported interaction is not a robust or generalizable finding.

## BACKGROUND

In recent years scientists in fields ranging from biomedicine to psychology have become increasingly concerned with the difficulties of replicating findings that are often assumed to be universal and reproducible (*Cesario, 2014*; *Ferguson & Mann, 2014*; *Moonesinghe, Khoury & Janssens, 2007*). In particular, experimental psychologists have been concerned about the fragility of priming effects, highlighting the need for replication of priming experiments (*Cesario, 2014*; *Moonesinghe, Khoury & Janssens, 2007*). In this paper, we report our attempts to replicate a finding of particular interest to us, which has been influential and highly cited.

The study we sought to replicate was that of *Griskevicius et al. (2011b)*. Across three experiments, they found that, following exposure to a mortality-risk prime (a fake newspaper article about rising violent crime, designed to elicit the sense that the world is

Corresponding author
Gillian V. Pepper,
gillian.pepper@ncl.ac.uk

dangerous and unpredictable), participants who grew up wealthier took fewer risks in a lab-based risky choice task, while those who grew up poorer took more risks. The same mortality-risk prime led participants who grew up wealthier to prefer future rewards to immediate ones in a lab-based delay discounting task. Meanwhile, those who grew up in poorer environments preferred more immediate rewards after mortality priming. Thus, for both risk acceptance and delay discounting, there were interactions between priming condition and childhood socioeconomic status in predicting the outcome. Only the interactions were significant in the original study: there were no overall directional effects of either the prime, or childhood socioeconomic status. This finding of an interaction between acute mortality priming and childhood socioeconomic background has been widely cited. However, there have been no direct replications of the original experiments. The original study used student participants from a large university in the USA. Here, we report three replications using British samples, one sample of university students, and two online samples. As the three experiments reported in the original paper were extremely similar to one another, both in design and in outcome, we focussed on a single one (experiment 2), performing multiple replications to maximize the precision of our estimates of the effects. We chose to focus on replicating experiment 2 because the control condition did not use a prime. Using only one adapted prime, rather than two, meant that our replication was closer to the original study.

## METHODS

We carried out three experiments that replicated experiment 2 from *Griskevicius et al. (2011b)*. These studies were granted ethical approval by the Newcastle University Faculty of Medical Sciences ethics committee (reference number 00554). The pre-registered protocols are available online at https://osf.io/6ucmq/ and https://osf.io/drq98/. All aspects of the replications including the informed consent and debrief screens, demographic questions, prime presentation, and outcome measures (see 'Measures'), were presented in a web browser via Qualtrics (Qualtrics, Provo, UT, ©2016). Participants in the lab-based study (replication 1) received an additional verbal debrief.

As in *Griskevicius et al. (2011b)*, participants in all three experiments were told that the newspaper article they read in the mortality priming condition (see 'Priming material') was part of a memory test. They were told that, after reading the article, they would complete questionnaires related to financial preferences to allow for memory decay. Participants were randomly allocated to either the mortality-priming condition or the control condition by Qualtrics. In the control condition, there were no priming materials. Control participants simply indicated their preferences for the risky and delayed rewards (see 'Measures'). The order of presentation for the risky choice and delay discounting outcome measures was randomised in all conditions. Following this, participants in all conditions indicated their childhood socioeconomic status (SES) by answering the questions outlined below (see 'Measures').

**Table 1** The characteristics of our study samples, compared to those in experiment 2 of *Griskevicius et al. (2011b)*.

|  | Original experiment | UK replication 1 | UK replication 2 | UK replication 3 |
|---|---|---|---|---|
| *n* | 71 | 72 | 159 | 162 |
| Males, females | 36, 35 | 9, 63 | 85, 74 | 85, 77 |
| Mean age (sd) | 20.8 (nr) | 19.8 (2.0) | 38.9 (11.5) | 36.3 (11.9) |
| Mean child SES (sd) | nr | 15.4 (3.0) | 11.6 (4.3) | 11.7 (4.1) |
| Mean adult SES (sd) | nr | 13.3 (3.5) | 11.7 (4.6) | 12.5 (4.4) |
| Sample | University students, for course credit | University students, for course credit | Online participants, for money | Online participants, for money |

**Notes.**
sd, standard deviation; nr, not reported.

## Participants and recruitment

The experimental treatments and the variables collected (see 'Measures') were the same in all three of our experiments. The key differences between the three experiments were in the participant demographics, and the method of recruitment: For our first replication, which took place in a laboratory at Newcastle University, the participants were 72 undergraduate University students (mean age = 20, 9 male, 63 female, see Table 1). The sample size for this first replication aimed to match that of the original experiment, which had 71 participants. For our second replication, 159 participants (mean age = 39, 85 male, 74 female, Table 1) were recruited online via Crowdflower (http://www.crowdflower.com). Crowdflower is an internet crowdsourcing platform through which users can be paid to complete online tasks, including surveys and experiments (for a useful review of Crowdsourcing platforms as research tools, see *Peer et al., 2017*). For our third replication, 162 participants (mean age = 36, 85 male, 77 female, Table 1) were recruited via Crowdflower and, if allocated to the priming condition, were asked additional questions about the prime as a manipulation check (see sections on 'Priming material' and 'The effects of prime perception'). In both online experiments (replications 2 & 3), 200 participants were initially recruited. Responses from those participants were then subjected to a series of quality-control checks. The quality control checks were designed to ensure that participants were really from the UK, that they were not repeat participants, and to increase the likelihood that they had been adequately exposed to the prime. After these checks, we were left with 159 participants in replication 2, and 162 participants in replication 3 (Table 1).

The quality checks used were as follows: (1) that participant identification numbers matched on Qualtrics and Crowdflower and were unique (if participants had attempted to complete the study multiple times, only the data from their first attempts were included in our analyses); (2) that the participants took a reasonable time (established using timestamp data from our lab-based replication) to complete the study: data were excluded from participants who took less than 60 s (the minimum time needed to honestly complete the experiment in the control condition), or more than 15 min (a long completion time indicates that a participant may have been interrupted during the experiment, potentially allowing any priming effect to wear off); (3) that the participants completed the experiment from a device with a UK-based internet protocol (IP) address (our adapted priming material described violent crimes in the UK and so might not have been effective for participants

living in other countries), and that the IP addresses matched between the Crowdflower and Qualtrics platforms (a further verification that the location information was genuine); (4) that participants declared themselves to be current UK residents when explicitly asked; (5) that the participants had entered the correct verification code (generated by Qualtrics at the end of the experiment) via the Crowdflower platform.

### Priming material

The original experiment (2 of 3 by *Griskevicius et al., 2011b*), used two conditions—one mortality prime, and one control condition in which there was no prime. The original mortality prime was a fake New York Times story entitled "Dangerous Times Ahead: Life and Death in the 21st Century," describing violent trends in the USA. We adapted this newspaper story (provided by Griskevicius et al., pers. comm., 2012 & 2015) for a British audience, altering descriptions of shootings so that they were about stabbings (which are more plausible events in the UK) and mentioning terrorist attacks that had occurred in the UK, rather than in the USA. In doing so, we deleted 108 words from the original prime, and added 131 words. The adapted prime is available as part of our pre-registered protocol, which can be seen online at https://osf.io/6ucmq/.

We pilot tested the adapted prime with a sample of 23 British students (seven male, 16 female) in order to ensure that the adapted article had similar effects to the original. Our pilot participants came from a range of socioeconomic backgrounds, as measured by the Index of Multiple deprivation (IMD) ranks for their home postcodes, which ranged from 217 to 31,805 (from a possible range of 1–32,844). The IMD identifies deprived areas of the country by combining a range of economic and social indicators into a single score. These scores are considered a meaningful objective measure of socioeconomic status (*Danesh et al., 1999*; *McLennan et al., 2011*).

*Griskevicius et al. (2011a)* originally tested their prime for its effect on perceptions of safety, unpredictability and general arousal. We piloted our prime using the same questions, which were as follows: (1) "*To what extent did the story make you think the world will become a more dangerous place?*" (2) "*To what extent did the story make you think the world will become unsafe?*" (3) "*To what extent did the story make you think the world will become more unpredictable?*" (4) "*To what extent did the story make you think the world will become uncertain?*" (5) "*To what extent did the story make you feel more emotionally aroused?*" In addition to these questions, we added a question about how convincing our raters thought the article was: "*Did you find this article convincing?*" All of these prime piloting questions were answered on a 7-point likert scale, with 1 being "not at all" and 7 being "very much."

We found that our prime had a similar effect on general arousal and perceptions of danger to those found in the original prime piloting by *Griskevicius et al. (2011a)*. However, it had significantly less of an effect on perceptions of uncertainty (Table 2). Pilot participants found the article moderately convincing, the mean rating being 4.04 out of a possible 7.

### Measures

Participants in all three experiments were asked for their age, sex, and home postcode. Their postcodes were used to obtain Index of Multiple Deprivation (IMD) scores, which

**Table 2** Means, standard deviations (sd), and *t*-test results for the comparison between the results from piloting our prime and those of prime piloting in the original study by *Griskevicius et al. (2011a)*.

| Primed perception | Modified prime | | Original prime | | Prime comparison | |
|---|---|---|---|---|---|---|
| | Mean | sd | Mean | sd | t | p |
| Dangerous | 4.26 | 1.74 | 4.44 | 1.95 | −0.494 | 0.626 |
| Unsafe | 4.17 | 1.83 | 4.61 | 1.75 | −1.146 | 0.264 |
| Unpredictable | 3.87 | 1.87 | 4.74 | 1.71 | −2.237 | 0.036 |
| Uncertain | 3.65 | 1.70 | 5.04 | 1.22 | −3.926 | 0.001 |
| Arousal | 3.30 | 1.82 | 3.52 | 1.53 | −0.568 | 0.576 |
| Convincing | 4.04 | 1.99 | – | – | – | – |

are considered a good measure of socioeconomic status (SES) for people living in the UK (*Danesh et al., 1999*). This measure was not used by *Griskevicius et al. (2011b)*, but provided us with an objective measure of socioeconomic status, to be used alongside subjective childhood socioeconomic status (see Supplemental Information). As in the original study, childhood SES was measured by asking participants to rate their agreement (on a scale from 1 to 7) with the following statements: (a) "My family usually had enough money for things when I was growing up"; (b) "I grew up in a relatively wealthy neighbourhood"; (c) "I felt relatively wealthy compared to the other kids in my school". Agreement with these statements was then summed to create the childhood SES score. Subjective adult socioeconomic status was measured by asking participants to rate their agreement (on a scale from 1 to 7) with the following statements: (a) "I have enough money to buy things I want"; (b) "I don't need to worry too much about paying my bills"; (c) "I don't think I'll have to worry about money too much in the future." Again, participants' agreement with these statements was summed to create an adult SES score. The associations between these subjective scores and postcode-based deprivation scores for each replication are reported in Table S4.

We used the same outcome measures as the original study (*Griskevicius et al., 2011b*). After having read the fake newspaper article (mortality-priming condition), or not (control condition), participants answered questions designed to measure risk preferences or delay discounting (in randomised order). The risk preference questions were seven choices of the format, "*Do you want a 50% chance of getting £800 OR £____ for sure?*" with the certain amount increasing from £100 to £700 in £100 increments. For delay discounting, participants were offered seven choices structured as follows, "*Do you want to get £100 tomorrow OR £____ 90 days from now?*" with the delayed reward starting at £110 and increasing to £170 in £10 increments. As in the original study, a higher score on this measure indicates greater patience (less-steep discounting). For fuller details, see our protocol at https://osf.io/6ucmq/.

## Analysis

Analyses were carried out in R 3.1.3 using the ggplot (*Wickham, 2009*), dplyr (*Wickham et al., 2016*), car (*Fox et al., 2016*), psych (*Revelle, 2016*), metaphor (*Viechtbauer, 2010*), and pwr (*Champely et al., 2017*) packages. The data and R scripts used for the

analyses are available as Supplemental Information 1. The main analyses reported in this paper were general linear models including as predictors: the main effect of mortality-priming condition (prime or no prime); the main effect of childhood SES; and the interaction between mortality-priming condition and childhood SES. In addition, following *Griskevicius et al. (2011b)*, we performed a number of additional exploratory analyses in which adult SES and sex were added as additional predictors and 3-way interactions were explored. These extra analyses are reported in the Section S3. We report standard two-tailed significance tests. However, following the original analysis by *Griskevicius et al. (2011b)* we also calculated directed *p*-values ($p_{dir}$) for the critical interaction between condition and childhood SES. Directed tests are intended to enhance power relative to two-tailed significance tests, without ignoring the possibility of effects in the unexpected direction (*Rice & Gaines, 1994*).

Having completed our three replications, we meta-analysed them, using a random-effects meta-analysis model implemented in the 'metafor' package (*Viechtbauer, 2010*). This was to investigate the possibility that there might be small effects, not significant in any one of the replications considered individually, but detectable when the information from all three replications was combined.

At the end of our third replication, after the outcome measures had been recorded, we presented participants with the questions initially used for prime piloting, as a manipulation check (see 'Priming material'). We used general linear models to examine whether participants' responses to the primes predicted their risk or delay discounting responses in the priming condition, and whether they did so in interaction with childhood SES.

## RESULTS

### Individual replications

The results of the general linear models for our three replications are summarised in Table 3. The critical interaction was not significant in any of the experiments, either by two-tailed *p*-values or directed tests. The main effects were largely non-significant. Figures 1–3 reproduce for our three replications the plots used by *Griskevicius et al. (2011b)* to illustrate the interactions in their data.

### The effects of prime perception

To address the possibilities that we did not replicate the original findings in our first two attempts either because we had altered the prime, or because the prime was less credible to British participants, we collected additional data during replication 3: after the outcome variables had been recorded, participants in the mortality-priming condition were also presented with the prime-piloting questions, described under 'Priming material'.

We investigated whether participants' belief in the primes predicted their risk acceptance or delay discounting scores. In models controlling for childhood SES, the extent to which participants reported finding the prime convincing had no effect on their risk or delay discounting scores (risk $F_{1,83} = 0.17, p = 0.68$; discounting $F_{1,83} = 2.08, p = 0.15$). However, exploratory analyses revealed that participants who felt that the world was unsafe after reading the prime were also subsequently less willing to accept risky options

**Table 3** Results from the main general linear models for the three replications. $Df = 1, 68$ (replication 1); 1,155 (replication 2); 1,158 replication 3. $P$-values are two-tailed; $p_{dir}$ represents $p$-value from a directed test as described by *Griskevicius et al. (2011b)*.

| Replication | Predictor | F | p | B | SE(B) |
|---|---|---|---|---|---|
| *Replication 1* | *Risk acceptance* | | | | |
| | Condition | 1.78 | 0.19 | 0.43 | 0.32 |
| | Child SES | 0.57 | 0.45 | −0.20 | 0.26 |
| | Condition * Child SES | 0.77 | 0.38 ($p_{dir} = 0.24$) | 0.29 | 0.33 |
| | *Delay discounting* | | | | |
| | Condition | 0.03 | 0.87 | 0.09 | 0.50 |
| | Child SES | 0.24 | 0.62 | 0.20 | 0.40 |
| | Condition * Child SES | 0.10 | 0.76 ($p_{dir} = 0.78$) | −0.16 | 0.52 |
| *Replication 2* | *Risk acceptance* | | | | |
| | Condition | 2.53 | 0.11 | −0.40 | 0.25 |
| | Child SES | 3.40 | 0.07 | 0.30 | 0.16 |
| | Condition * Child SES | 0.03 | 0.87 ($p_{dir} = 0.70$) | −0.04 | 0.26 |
| | *Delay discounting* | | | | |
| | Condition | 2.03 | 0.16 | −0.54 | 0.38 |
| | Child SES | 0.81 | 0.37 | −0.22 | 0.24 |
| | Condition * Child SES | 1.82 | 0.18 ($p_{dir} = 0.11$) | 0.52 | 0.38 |
| *Replication 3* | *Risk acceptance* | | | | |
| | Condition | 1.53 | 0.22 | 0.35 | 0.28 |
| | Child SES | 5.26 | 0.02 | 0.45 | 0.19 |
| | Condition * Child SES | 0.59 | 0.44 ($p_{dir} = 0.97$) | −0.22 | 0.28 |
| | *Delay discounting* | | | | |
| | Condition | 0.10 | 0.75 | −0.13 | 0.42 |
| | Child SES | 1.72 | 0.19 | 0.38 | 0.29 |
| | Condition * Child SES | 0.01 | 0.93 ($p_{dir} = 0.58$) | 0.04 | 0.43 |

($F_{1,79} = 6.37, p = 0.01$). There was also a trend in which participants who reported feeling that the world would become more dangerous after reading the prime also discounted future rewards less steeply ($F_{1,79} = 3.23, p = 0.07$). Thus, we ran models testing for interactions between childhood SES and post-prime perceptions that the world was unsafe or dangerous—to test whether interaction effects with childhood SES would be visible in those participants that had been more-successfully primed with a sense of danger. The main effects of primed threat perceptions on risk and delay discounting remained significant in these models. However, there were no interactions between childhood SES and primed perceptions for risk and delay discounting (Table 4).

## Meta-analysis

When we meta-analysed the findings of the three experiments, the 95% confidence intervals for the parameter estimates overlapped zero in most cases, for both main effects and interactions (Fig. 4). Thus, we did not detect interaction effects in the individual replications, or when all three replications were combined.

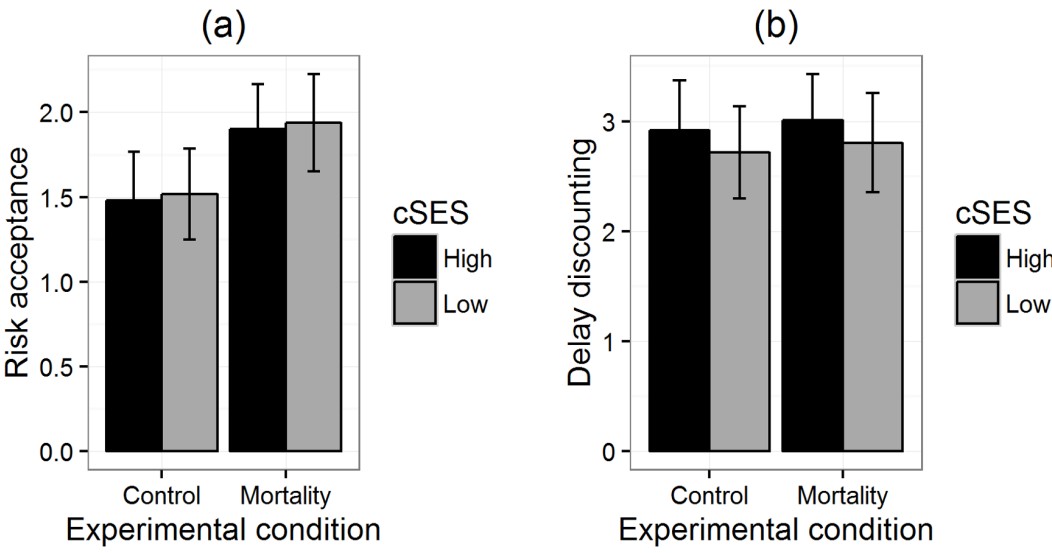

**Figure 1** **Risk acceptance (A) and delay discounting (B) by priming condition for participants of high and low childhood SES in replication 1.** Error bars represent one standard error.

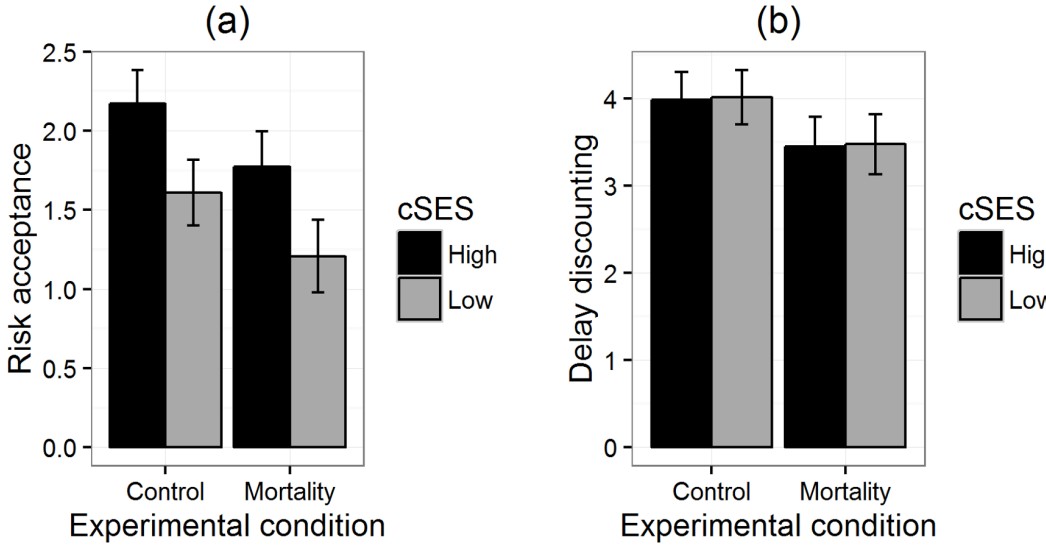

**Figure 2** **Risk acceptance (A) and delay discounting (B) by priming condition for participants of high and low childhood SES in replication 2.** Error bars represent one standard error.

## Additional analyses

In the additional analyses reported in the Supplemental Information 2, the only recurrent finding was a trend towards an interaction between sex and mortality-priming condition in predicting delay discounting. This interaction was marginally non-significant in each individual replication, but was significant in a meta-analysis across all three replications ($B = 1.66$, s.e.(B) $= 0.55$, $z = 3.00$, $p < 0.01$). After priming, men became more patient (that is, discounted the future less steeply), whereas women became less patient (that is, discounted more steeply; see Figs. S1 and S2).

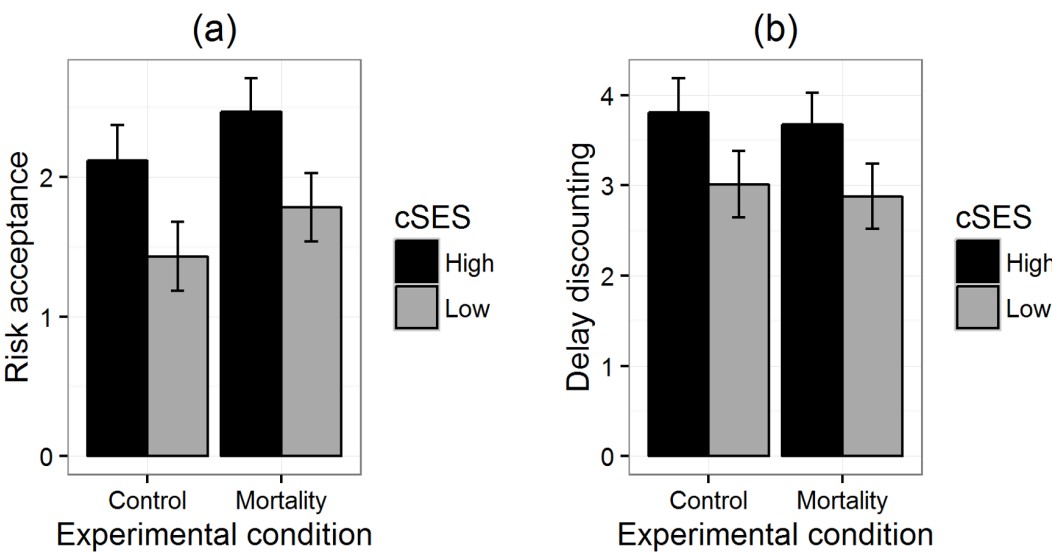

**Figure 3  Risk acceptance (A) and delay discounting (B) by priming condition for participants of high and low childhood SES in replication 3.** Error bars represent one standard error.

**Table 4  Results of the model examining interaction effects for primed danger perceptions and child SES scores on delay discounting in replication 3.** $Df = 1, 82$. "Unsafe" refers to participants responses to the question "To what extent did the story make you think the world will become unsafe?", and "Dangerous" refers to participants responses to the question "To what extent did the story make you think the world will become a more dangerous place?"

| Risk acceptance | F | p | B | SE(B) |
|---|---|---|---|---|
| Unsafe | 5.61 | 0.02 | −0.37 | 0.16 |
| Child SES | 0.85 | 0.36 | 0.75 | 0.82 |
| Unsafe * Child SES | 0.43 | 0.51 | −0.10 | 0.15 |

| Delay discounting | F | p | B | SE(B) |
|---|---|---|---|---|
| Dangerous | 4.73 | 0.03 | 0.45 | 0.21 |
| Child SES | 0.20 | 0.66 | 0.41 | 0.92 |
| Dangerous * Child SES | 0.03 | 0.86 | −0.03 | 0.17 |

## DISCUSSION

We have reported three attempts to replicate findings by *Griskevicius et al. (2011b)*, who found consistent interaction effects between childhood SES and mortality-priming condition for two outcomes: risk acceptance and delay discounting. We did not replicate this main finding in our three experiments which, unlike the original study, used British participants. In none of our individual replications was the predicted interaction significantly different from zero. We found no evidence of an interaction when we combined the results of the three replications in a meta-analysis. Our samples were diverse (one student, two online), and contained a reasonable degree of variation in childhood SES, as would have been necessary to reveal an interaction. We cannot directly compare the childhood SES variation in our sample with that in the sample of *Griskevicius et al. (2011b)*,

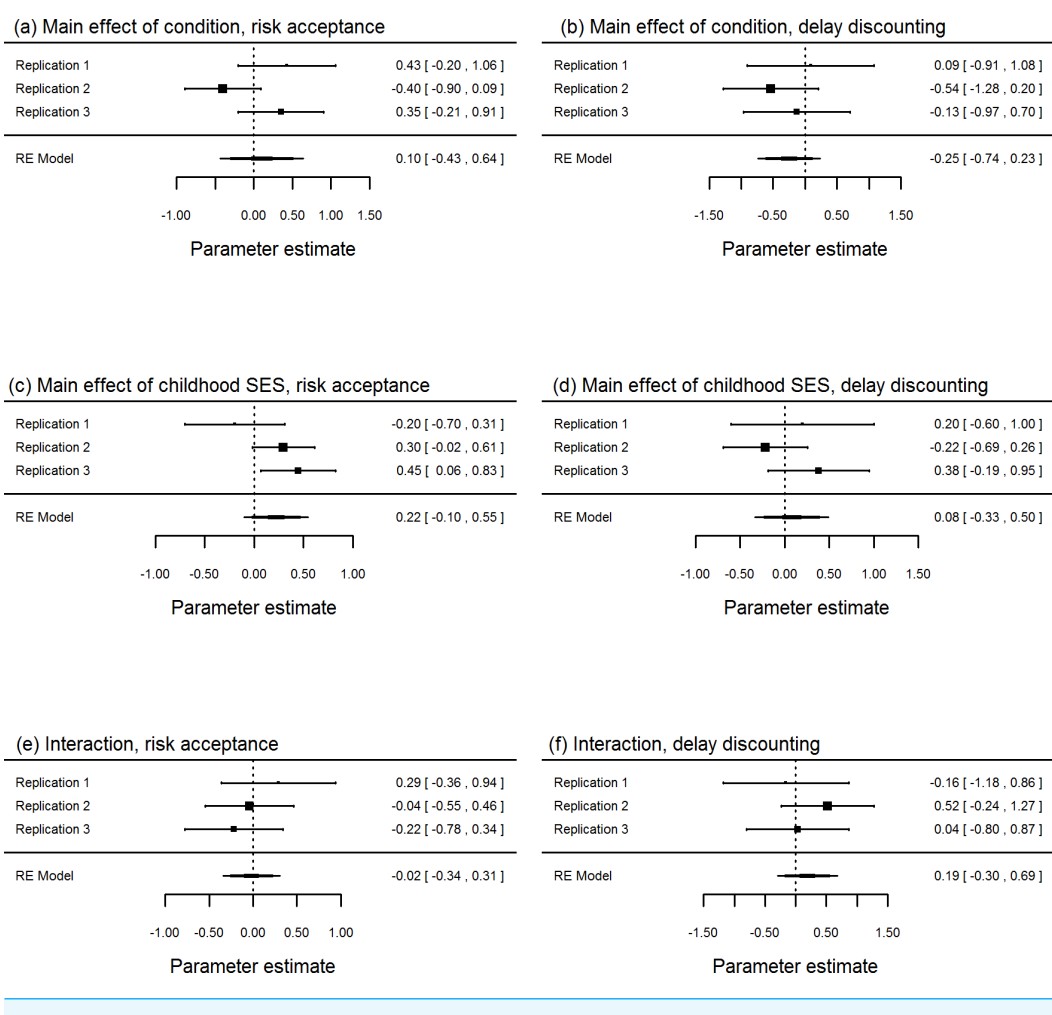

**Figure 4** **Forest plots from meta-analyses across our three experiments, showing the main effects of mortality priming condition (A, B); the main effects of childhood SES (C, D), and interaction between mortality-priming condition and childhood SES (E, F).** Shown are the central estimates of effect size, and the 95% confidence intervals.

as they do not report descriptive statistics, but it seems unlikely our samples contained much less variation.

There are a number of potential reasons for our failure to replicate the original findings of *Griskevicius et al. (2011b)*. Firstly, we used British undergraduate students (replication 1) and online participants from the more general British population (replications 2 & 3), whilst the original study used a North American undergraduate sample. Thus, demographic and cultural differences could explain differences in either our participants' behavioural responses to the priming material, or their willingness to believe the information given in the prime. Indeed, we altered the original priming material in order to make it more convincing to a British audience (for example by replacing references to gun violence with references to knife attacks—see 'Priming material'). Using the same piloting questions as Griskevicius et al., we found that our prime had similar effects on perceived danger (dangerous, unsafe: Table 2) in its respective

audience to the original prime. However, our prime had significantly less effect on perceived uncertainty (uncertain, unpredictable: Table 2). In addition, we asked piloting participants how convincing they found the priming article. In both the initial pilot (Table 2), and in replication 3 (Table S13), participants were only moderately convinced by the priming information (4–5 points on a 7-point scale, with a score of 7 signifying that the prime was very convincing). *Griskevicius et al. (2011a)* did not ask their pilot participants to rate the prime for convincingness, and so we are unable to directly compare, but it is possible that our participants found our version of the prime less convincing than participants in the original experiments.

Our replications were also limited in other ways. Although our first replication matched the original study sample size ($n_1 = 72$, Table 1), it may well have been underpowered. Indeed, power analyses indicated that, assuming 80% power and a significance level of 0.05, replication 1 was powered to detect a minimum detectable main effect (MDE) of 0.11. This is a small-to-medium effect according to convention (*Champely et al., 2017*). Replications 2 and 3 had much greater power ($n_2 = 159$, $MDE_2 = 0.05$, $n_3 = 162$, $MDE_3 = 0.05$), but used an online crowdsourcing platform for participant recruitment. We applied rigorous quality controls (see 'Participants and recruitment') to ensure that the data came from participants living in the UK (the target audience for the modified prime), and that the participants didn't take too long (an indicator that priming may have been interrupted) or too short (an indicator that the participants may have answered questions without reading them thoroughly) a time to complete the study. Nonetheless, we had no control over the environments in which the online participants experienced the experiment—something which may have affected the efficacy of priming. Finally, crowdsourcing platforms have limited systems in place to prevent their users "cheating" (e.g., by taking part in a study twice, *Peer et al., 2017*). We tried to reduce this problem by excluding repeat attempts from the same IP address (see 'Participants and recruitment'). However, this may not be sufficient to prevent a few users repeatedly participating from different devices.

Across our three experiments, we saw some evidence that participants from higher-childhood SES backgrounds were more accepting of risk. This result was marginally non-significant overall. However, the associations were stronger in the two non-student samples, where the variation in childhood SES was larger. We also found, for delay discounting, an interaction between sex and priming that was significant across the three studies in meta-analysis: men tended to respond to the prime by becoming more patient, whereas women tended to become less patient (see Figs. S1 & S2). As this effect was not the subject of an a priori prediction, and was not reported by *Griskevicius et al. (2011b)*, we interpret it with caution. We found no evidence of a three-way interaction between prime, sex, and childhood SES (see Section S3), and thus there is no reason to believe that differing sex balances in our samples and those of *Griskevicius et al. (2011b)* explain this difference in results.

Although it is possible that cross-cultural, demographic or methodological differences might account for our failure to replicate the original finding, it is also possible that the original findings were a false positive. Set against this possibility is that fact that *Griskevicius et al. (2011b)* reported three similar experiments using the same materials, and obtained

the same result each time. We note that one attempt at conceptually replicating the effect in US undergraduate students also found no evidence of the interaction (*Frederick, Khan & Ancona, 2016*). However, this study by Frederick et al. used different priming material and, as the authors note, priming effects seem to be particularly sensitive to methodological differences (*Cesario, 2014*). Thus we suggest that further attempts at replication, both in the US population and in other populations internationally, are needed.

## ACKNOWLEDGEMENTS

We are grateful to our participants for providing data, and to Katherine Corker and one other anonymous reviewer for their helpful comments and suggestions.

### Funding

This work was funded by European Research Council grant AdG 666669 - COMSTAR. The funders had no role in study design, data collection and analysis, decision to publish, or preparation of the manuscript.

### Grant Disclosures

The following grant information was disclosed by the authors:
European Research Council: AdG 666669 - COMSTAR.

### Competing Interests

The authors declare there are no competing interests.

### Author Contributions

- Gillian V. Pepper performed the experiments, analyzed the data, contributed reagents/materials/analysis tools, wrote the paper, prepared figures and/or tables, reviewed drafts of the paper.
- D Helen Corby performed the experiments, analyzed the data, contributed reagents/materials/analysis tools, reviewed drafts of the paper.
- Rachel Bamber, Holly Smith and Nicky Wong performed the experiments, contributed reagents/materials/analysis tools, reviewed drafts of the paper.
- Daniel Nettle analyzed the data, contributed reagents/materials/analysis tools, wrote the paper, prepared figures and/or tables, reviewed drafts of the paper.

### Human Ethics

The following information was supplied relating to ethical approvals (i.e., approving body and any reference numbers):

These studies were granted ethical approval by the Newcastle University Faculty of Medical Sciences ethics committee (reference number 00554).

### Supplemental Information

Supplemental information for this article can be found online at http://dx.doi.org/10.7717/peerj.3580#supplemental-information.

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
