# Peer review of "The influence of mortality and socioeconomic status on risk and delayed rewards: a replication with British participants"

_PeerJ, doi:10.7717/peerj.3580_

## Round 0.1 · original submission · Major Revisions

Both expert reviewers were impressed with your manuscript, but each identified several issues that must be addressed before it is suitable for publication. I believe that all of the issues they raised require changes to the manuscript, although you are free to express disagreement in your rebuttal letter.

Reviewer 1 ·

Basic reporting

The English was clear, but overly wordy and repetitive. For example, Line 50 can be shortened to: We carried out three experiments that replicated experiment 2 from Griskevicius et al (2011). As currently written it reads more like an essay than a concise research article.

Relatively little literature was cited. However, the cited literature was well chosen.

The structure is as expected. However, the tables are somewhat odd looking with the grey lines vs. bold black lines. This is not what I would expect to see in a journal. Table 2, the left column would benefit from a descriptive column heading. In Table 4, what does Unsafe versus Dangerous refer to? The figures need more informative axis labels. e.g. "risk taking" and "delay discounting" on the y-axes. On the x-axis "priming condition" is an inaccurate axis label. The control is not a prime. The control is went straight to survey without reading anything. "Experimental condition" is more accurate. Many readers just glance at tables and figures without reading the text. It is helpful is these are informative and somewhat self-contained.

The article is self-contained. The results are relevant to the hypotheses. However, I am left wanting more. Substantially more. See, validity of findings for a discussion of my issues with this.

Other issues:
Line 23: Given this journal has a wide audience, are the authors speaking generally or just about experimental psychology research?

Experimental design

The research is within the aims and scopes of PeerJ.

This research question is needed and is timely. It will be of primary interest to researchers within Experimental Psychology, and the relatively small group of people who have used Griskevicius' prime. However, the replication of psychology research has been discussed in numerous prestigious and wide reaching journals recently. Meaning a larger audience that expected will likely be interested in this article. Furthermore, psychology research is often of interest to the general public and more likely to be reported in the popular press.

The authors replicated the original study relatively well. With only necessary modifications. However, they failed to address a primary issue with the original work, the small sample size. Also, the authors may not have replicate the +/- 1 SD from the mean for childhood SES parts of the original. This research meets ethical standards.

Methods well described and could be replicated. Also, the authors should be lauded for pre-registering their hypotheses and methods. A major concern with this type of research is p-hacking. The pre-registration of the work in this paper adds credence and strength to their results.

Validity of the findings

This article barely touches on the numerous issues with Griskevicius et al (2011), and has some similar issues. For example, sample size, as with Griskevicius et al (2011) is small. My main issue, with that article, and now this article, is the small sample size. With sex and other variables in the model, what is the power to detect an effect with such a small sample size? Moreover, in the original article, an effect was only seen when looking at people +/- one standard deviation from the mean for childhood SES. It is unclear to me whether the current paper also looks at this (an issue which needs addressing). If it does, what credence can be given to any results considering the even smaller sample sizes once only looking at +/- one SD from the mean (i.e. 42% of the original sample size). Force positives and negatives should be expected under these circumstances. A more meaningful replication study would have a large and robust sample size, and, thus, reduce the risk of force positives or negatives.

Recruiting from non-university sources is great. The authors should be lauded for this. However, again the sample size remains small, and why expect a linear relationship between childhood SES and risk taking or delay discounting? Thresholds may exist, e.g. Childhood SES must be below some value for a person to have a different underlying life history strategy and thus respond differently to mortality cues than others. Perhaps in high-income countries even people with seeming low childhood SES are above this threshold as far as the underlying, evolved psychological mechanisms are concerned? Some discussion of theory would be good. Do they disagree solely with the methods or statistics of the original article, or is the underlying theory and conclusions what troubles them? This needs to be addressed. Personally, some discussion of theory and whether the original results and their own are consistent with this would be appreciated.

Additional comments

Line 261: Among US undergraduates at the University where the original study was run, it is not uncommon to read the New York Times. The original article would never be published in the New York Times. I would imagine very few students considered it to be real. Furthermore, most students taking the survey were doing so for course credit. This was most likely for Introduction to Psychology, meaning they probably could guess the point of the study to some degree, and were very unlikely to think the article real. A deeper discussion of this issue would highlight the importance of also working in non-student populations and argue for why they recruited via CrowdFlower non-students.

·

Basic reporting

See below.

Experimental design

See below.

Validity of the findings

See below.

Additional comments

The current paper reports three direct replications of Griskevicius et al. (2011) Experiment 2. This paper has been frequently cited (over 250 times on Google Scholar). I reviewed the manuscript itself, the pre-registration, and the data files/code submitted with the manuscript. I also reviewed the original study to check closeness of the replications. Overall the replications appear to be well done and true to the original. I have a few points to raise and a few questions for the authors to address, however.

1. Validity of the mortality salience manipulation: I very much appreciate that the authors pilot tested their modified priming news article to ensure that it was comparable to the original study. On the same five item questionnaire that the original authors used, the current authors get similar ratings on three of five items. There are a couple of things to note. First, the authors cite Grivkevicius et al. (2011) for the original pilot study of these materials, but the correct citation is Grivkevicius et al. (2010). Second, it appears that the same control condition materials were used in the replication as in the original (can you please make that clear in the manuscript?), but the control materials were not pilot tested. However, by going to Grivkevicius et al. (2010), we can compare the current data to the control condition (not in Table 2) as well as the experimental condition (already in Table 2). There we see that for unpredictable and uncertain that the current means are still much higher than the old control condition, even though they are lower than the old experimental condition. The difference in means is over one full scale point in both cases. Thus, even though the modified story is not identical to the original, it still seems to invoke many of the same features as the original. As an aside, your tables and figures are very well done, and I appreciated all the supplements. Is it possible to post also the original passage from Grivkevicius et al. 2010/2011? I realize you might have to get permission to share, but it would be valuable to have both passages on OSF.

2. Type I vs. Type III Sums of Squares in R: In reviewing the authors’ R code (which was very well done and extremely clear, by the way), I noticed that models were run two ways: using the anova() command and using the lm() command. It is a little known quirk of R that by default anova() uses Type I sums of squares, whereas lm() uses Type III. SPSS and SAS use type III by default. Results under the two procedures will therefore be similar but not identical (for example, in your first analysis for replication one, the p value for your condition variable will either be .193 (using anova) or .187 (using lm). What follows is code to run this first ANOVA (and get correct F statistics) under Type III sums of squares:

Anova(RiskInteractionMain1,contrasts=list(topic=contr.sum, sys=contr.sum), type=3)

Your conclusions won’t change, but the F and p values in Table 3 need to be corrected.

3. Directional tests: Related to my previous point, the p values for the directional tests may also need to be corrected (though I see in the code that they come from the lm() command, so perhaps they are OK). However, I did wonder whether you did the directional tests correctly. If the regression coefficient for the interaction is positive, your code produces a p-value that is higher than the two-tailed p, and if the regression coefficient for the interaction is negative, your code produces a p-value that is lower than the two-tailed p. This implies that the hypothesized interaction is negative. Frankly, I can’t be sure which direction the hypothesized interaction would be, and I find the whole logic behind repartitioning the type I error to be somewhat bizarre in the case of an interaction term. (I do appreciate that you tried to recreate the original authors’ somewhat unusual decision.) In any event, if you have done the directional tests backwards, to reverse them you would want to use this code (for analysis 1, for instance):

p1tRisk1 = pt(TValueRisk1, df=dfRisk1,lower.tail=F)

4. Power: Each of these replication studies are likely underpowered to detect the key interaction, but the fact that three attempts have been made likely puts the total power at a more acceptable level. It would be good to report the smallest effect that the current study is 80% powered to detect in the manuscript.

5. You chose to focus on Study 2 for the replication, but it doesn’t say why anywhere in the paper. Could you add this to the introduction?

6. Meta-analysis: I appreciate that the authors used meta-analysis to combine their replication studies. I’m of a mixed mind when it comes to including non-pre-registered studies in such an analysis, but it might make for an interesting follow-up to add the three original studies to the meta-analysis. Likewise if there are other similar studies, can they be added? I think the main effect of SES on risk is interesting – is there other literature on this topic you could look at? Maybe this goes too far afield for the current paper, but I thought I would suggest it anyway.

---

## Round 0.2 · accepted · Accept

Congrats - I think this is an outstanding contribution to the literature.

Reviewer 1 ·

Basic reporting

Some of the language remains more conversational than professional. For example, Line 35 – grew up wealthier and poorer reads as unprofessional terminology. Especially as there is a sudden change to saying adding “environment” to the end in line 39.

Otherwise the article meets all criteria.

Experimental design

All criteria met or exceeded.

Validity of the findings

I am concerned that the original paper did slightly different statistics than done here. However, that is through no fault of the authors. The original paper is extremely unclear in how it did the interaction term and what else may have been done. It is only from a personal communication with one of the authors of the original paper that I know about the statistics in some detail. Here is the relevant part of this personal communication "childhood SES was a continuous variable in each of our studies, and point estimates of the prime effect at one standard deviation above (i.e., high SES) and below (i.e., low SES) the mean were tested." If the authors of the article under review did this point estimate, please make it clearer in the text. However, as no significant interaction term was found by in the paper under review this may be a moot point, but perhaps one the authors should still mention?

Additional comments

Authors addressed all concerns of both myself and the other reviewer. I still argue that a larger sample size would be more telling, but understand the difficulty in get this.

·

Basic reporting

No comment

Experimental design

No comment

Validity of the findings

No comment

Additional comments

The authors have done a nice job addressing all of my previous concerns. I am happy to endorse this version of the manuscript.